# Targeted intracellular degradation of SARS-CoV-2 via computationally optimized peptide fusions

Pranam Chatterjee [1,2,3,4✉], Manvitha Ponnapati[1,2], Christian Kramme[3,4], Alexandru M. Plesa [3,4], George M. Church[3,4] & Joseph M. Jacobson[1,2]

The COVID-19 pandemic, caused by the novel coronavirus SARS-CoV-2, has elicited a global health crisis of catastrophic proportions. With only a few vaccines approved for early or limited use, there is a critical need for effective antiviral strategies. In this study, we report a unique antiviral platform, through computational design of ACE2-derived peptides which both target the viral spike protein receptor binding domain (RBD) and recruit E3 ubiquitin ligases for subsequent intracellular degradation of SARS-CoV-2 in the proteasome. Our engineered peptide fusions demonstrate robust RBD degradation capabilities in human cells and are capable of inhibiting infection-competent viral production, thus prompting their further experimental characterization and therapeutic development.

---

[1] The MIT Center for Bits and Atoms, Cambridge, MA 02139, USA. [2] MIT Media Lab, Massachusetts Institute of Technology (MIT), Cambridge, MA 02139-4307, USA. [3] Department of Genetics, Harvard Medical School, Boston, MA 02115, USA. [4] Wyss Institute for Biologically Inspired Engineering, Harvard University, Center for Life Science Bldg., Boston, MA 02115, USA. ✉email: pranam@mit.edu

 1

SARS-CoV-2 has emerged as a highly pathogenic coronavirus that has now spread to over 200 countries, infecting ~50 million people worldwide and killing over 1 million as of October 2020[1]. Economies have crashed, travel restrictions have been imposed, and public gatherings have been canceled, all while a sizeable portion of the human population remains quarantined. Rapid transmission dynamics as well as a wide range of symptoms, from a simple dry cough to pneumonia and death, are common characteristics of coronavirus disease 2019 (COVID-19)[2]. With no cures readily available[3], and only five vaccines approved for early or limited use (in China and Russia, specifically)[4], there is a pressing need for robust and effective therapeutics targeting the virus.

Numerous antiviral strategies have been proposed to limit SARS-CoV-2 replication by preventing viral infection and synthesis[5]. As SARS-CoV-2 is a positive-sense RNA virus, Abbott, et al. recently devised a CRISPR-Cas13d based strategy, termed PAC-MAN, to simultaneously degrade the positive-sense genome and viral mRNAs[6]. While this method may serve as a potential prophylactic treatment, introducing foreign and relatively large components such as Cas13 enzymes into human cells in vivo presents various delivery and safety challenges[7].

The most rapid and acute method of protein degradation intracellularly is at the post-translational level. Specifically, E3 ubiquitin ligases can tag endogenous proteins for subsequent degradation in the proteasome[8]. Thus, we hypothesize that by employing guiding E3 ubiquitin ligases with viral-targeting peptides, one can mediate depletion of SARS-CoV-2 viral components in vitro.

In this study, we devise a targeted intracellular degradation strategy for SARS-CoV-2 by computationally designing peptides that bind to its spike (S) protein receptor binding domain (RBD) and recruit a human E3 ubiquitin ligase for subsequent proteasomal degradation. Our experimental results identify an optimal peptide variant that mediates robust degradation of the RBD fused to a stable superfolder-green fluorescent protein (sfGFP)[9] in human cells and inhibits infection-competent viral production, thus motivating further exploration of this strategy from a therapeutic perspective.

## Results

### Computationally optimized peptides targeting the SARS-CoV-2 RBD.
Since the 2003 SARS epidemic, it has been widely known that the angiotensin-converting enzyme 2 (ACE2) receptor is critical for SARS-CoV entry into host cells[10]. ACE2 is a monocarboxypeptidase, widely known for cleaving various peptides within the renin-angiotensin system[11]. Functionally, there are two forms of ACE2. The full-length ACE2 contains a structural transmembrane domain, which anchors its extracellular domain to the plasma membrane. The extracellular domain has been demonstrated as a receptor for the S protein of SARS-CoV, and recently, for that of SARS-CoV-2[12]. The soluble form of ACE2 lacks the membrane anchor, thus preserving binding capacity, and circulates in small amounts in the blood[13].

Recently, it has been shown that soluble ACE2 (sACE2) can serve as a competitive interceptor of SARS-CoV-2 and other coronaviruses by preventing binding of the viral particle to the endogenous ACE2 transmembrane protein, and thus viral entry[14]. sACE2, however, is capable of binding other biological molecules in vivo, most notably integrin receptors[15]. It is critical to ensure therapeutics targeting SARS-CoV-2 epitopes withstand the possibility of viral mutation, which may allow the virus to overcome the host adaptive immune response[16]. We thus conducted in silico protein modeling to engineer minimal sACE2 peptides that not only maintain potent RBD binding, but also possess reduced off-target interaction with the integrin $\alpha 5\beta 1$ receptor. Finally, it has been demonstrated that the SARS-CoV-2 S protein is highly resistant to mutation, with 99.8% of the residues being conserved across multiple strains. More specifically, the S-RBD contains mostly conserved and highly conserved residues, with only minimal amino-acid changes, most of which possess equivalent properties[17–19]. Thus, we assessed whether our RBD-protein targeting peptides exhibit cross-binding affinity toward previous spike proteins for which a known structure exists, thus allowing us to determine their tolerance to viral evolution (Fig. 1a).

To do this, we retrieved a structure of the SARS-CoV-2 RBD bound to sACE2 from the Protein Data Bank (PDB 6M0J)[20]. We first utilized the PeptiDerive protocol[21] in the Rosetta protein modeling software[22] to generate truncated linear sACE2 peptide segments between 10 and 150 amino acids with significant binding energy compared to that of the full SARS-CoV-2 RBD-sACE2 interaction. To analyze the conformational entropy of the peptide segments in the binding pocket, we employed both the FlexPepDock and Protein–Protein protocols[23] to dock the peptides to the original RBD (Fig. 1b.i). To ensure tolerance to potential mutations in the RBD, we docked peptides with optimal binding energies against the divergent 2003 SARS-CoV RBD bound with ACE2 (PDB 2AJF)[24] (Fig. 1b.ii). Peptides that demonstrated highest binding energy for SARS-CoV and SARS-CoV-2 RBD were then docked against the $\alpha 5\beta 1$ integrin ectodomain (PDB 3VI4)[25] to identify weak off-target binders (Fig. 1b.iii). Finally, as bounded ligands may alter the native conformation of proteins, we confirmed the conformational stability of the candidate peptides as monomers to rule out any destabilizing factors in their unbound state. After applying these filters, 26 candidate peptides were selected from a total list of 188 initial peptides (Fig. 1c).

### Targeted degradation of RBD with TRIM21.
TRIM21 is an E3 ubiquitin ligase that binds with high affinity to the Fc domain of antibodies and recruits the ubiquitin-proteasome system to degrade targeted proteins[26]. Recently, the Trim-Away technique was developed for acute and rapid degradation of endogenous proteins, by co-expressing TRIM21 with an anti-target antibody[27]. We thus hypothesized that by fusing the Fc domain to the C-terminus of candidate peptides and co-expressing TRIM21, we can mediate degradation of the RBD fused to a stable fluorescent marker, such as superfolder GFP[9] (RBD-sfGFP), in human HEK293T cells using a simple plasmid-based assay (Fig. 2a, b). We chose the two most compact candidate peptides, an 18-mer and 23-mer derived from the ACE2 peptidase domain $\alpha 1$ helix, which is composed entirely of proteinogenic amino acids, as well as the candidate peptide computationally predicted to have highest binding affinity to the RBD (a 148-mer), for testing alongside sACE2 (Fig. 1c). We also tested a recently-engineered 23-mer peptide from Zhang, et al., purporting to have strong RBD-binding capabilities[28]. Five days post transfection, we analyzed the degradation of the RBD-sfGFP complex by flow cytometry. After confirming negligible baseline depletion of GFP+ signal with and without exogenous TRIM21 expression, as well as no off-target degradation of sfGFP unbound to the RBD, we observed over 30% reduction of GFP+ cells treated with full-length sACE2 fused to Fc and co-expressed with TRIM21, as compared to the RBD-sfGFP-only control. Of the tested peptides, only the 23-mer demonstrated comparable levels of degradation, with ~20% reduction in GFP+ cells (Fig. 2c).

### Engineering of an optimal peptide-based degradation architecture.
Recently, deep mutational scans have been conducted on

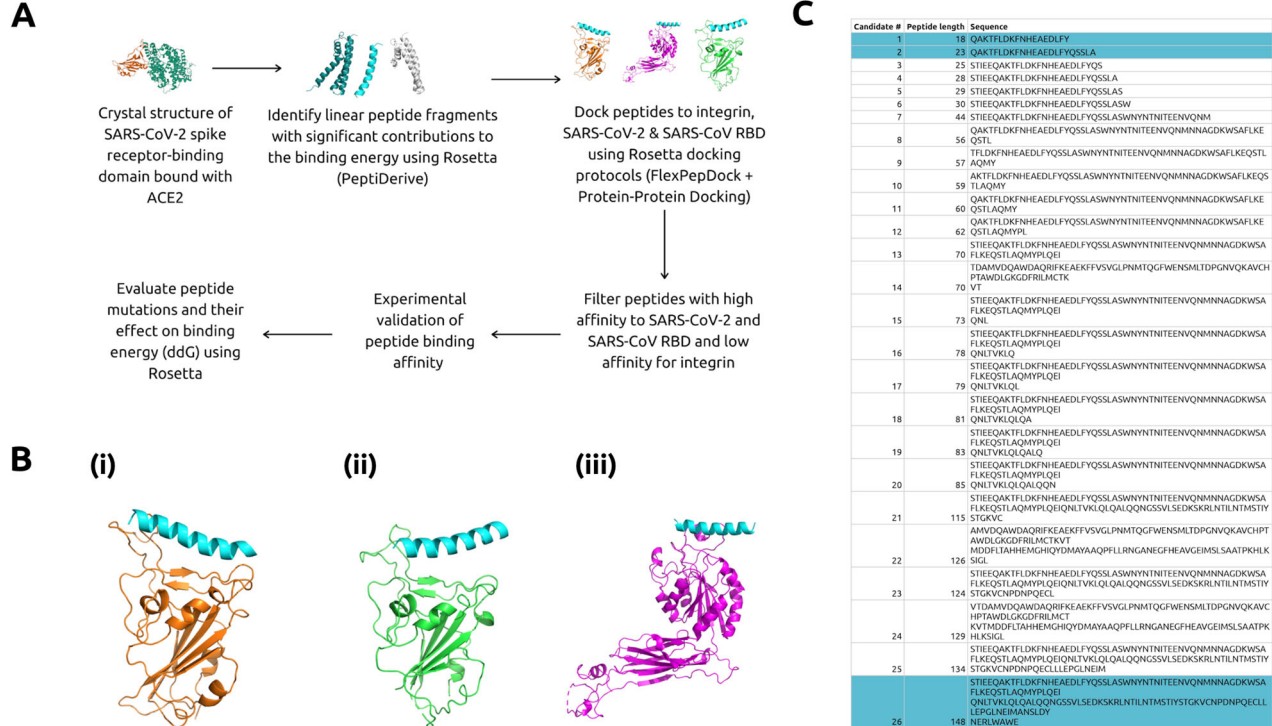

**Fig. 1 In silico design of RBD-targeting peptides. a** Flow chart detailing computational pipeline to obtain optimized peptides. **b** 23-mer peptide computationally docked to (i) SARS-CoV-2 RBD[20], (ii) SARS-CoV RBD[24], and (iii) integrin $\alpha5\beta1$ receptor[25] in Rosetta and visualized using PyMol. The 23-mer peptide is shaded in blue. **c** Candidate peptides selected after application of three filter docking steps. Peptides highlighted in blue indicate those chosen for experimental validation.

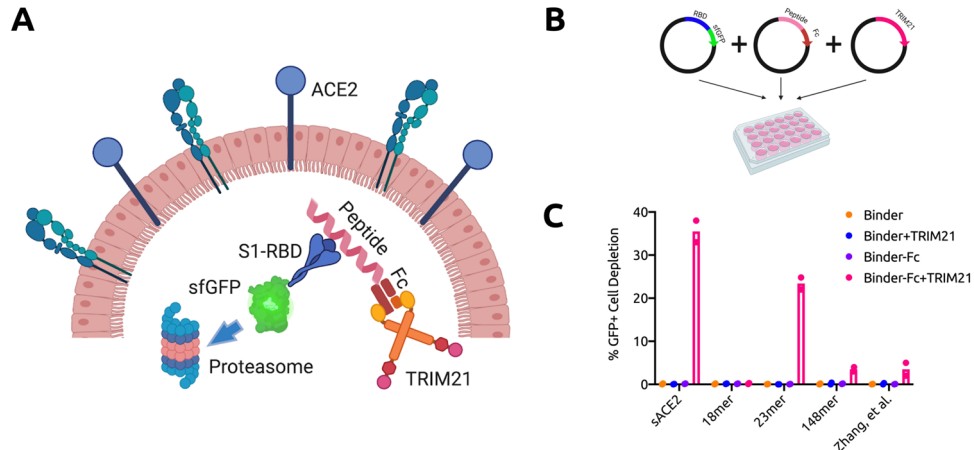

**Fig. 2 TRIM21-mediated degradation of RBD via peptide targeting. a** Architecture and mechanism of TRIM21-based degradation system. The Fc domain is fused to the C-terminus of RBD-targeting peptides. TRIM21 recognizes the Fc domain and tags RBD-sfGFP complexes for ubiquitin-mediated degradation in the proteasome. **b** Three plasmid assay used to experimentally validate degradation architecture in human HEK293T cells. All CDS are inserted into the pcDNA3.1 backbone. **c** Analysis of RBD-sfGFP degradation by flow cytometry, in the absence or presence of Fc (in cis), TRIM21 (in trans), or both. All samples were performed in independent transfection duplicates ($n = 2$) and gated on GFP+ fluorescence. Mean percentage of GFP+ cell depletion was calculated in comparison to the RBD-sfGFP only control.

sACE2 to identify variants with higher binding affinity to the RBD of SARS-CoV-2[29]. Similarly, we conducted a complete single point mutational scan for all 23 positions in the peptide using the ddG-backrub script in Rosetta to identify mutants with improved binding affinity[30]. For each mutation, 30,000 backrub trials were performed to sample conformational diversity. The top eight mutations predicted by this protocol were chosen for the experimental assay (Fig. 3a.i), along with the top eight mutations predicted using an Rosetta energy function optimized for predicting the effect of mutations on protein–protein binding[30] (Fig. 3a.ii), as well as the top eight mutational sites within the 23-mer sequence from deep mutational scans of sACE2[29] (Fig. 3a.iii). Our results in the subsequent TRIM21 assay identified A2N, derived from the original Rosetta energy function, as the optimal mutation in the 23-mer peptide, which achieved over 50% depletion of GFP+ cells, improving on both the sACE2 and 23-mer architecture as well as that of a previously optimized full-length mutant, sACE2v2.4 (Fig. 3b).

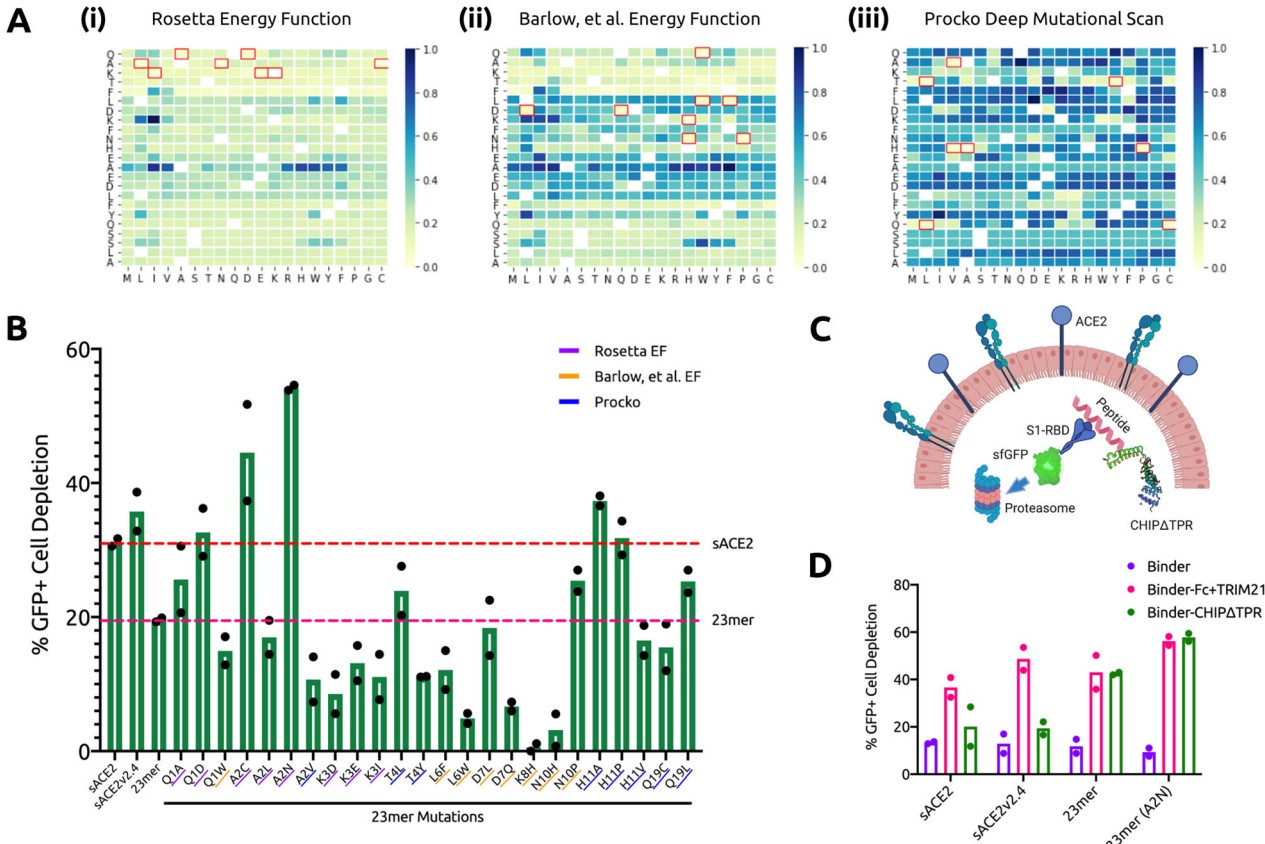

**Fig. 3 Engineering of optimal RBD degradation architecture. a** (i) ddG of 23-mer mutations predicted by Rosetta with the default energy function, (ii) ddG of 23-mer mutations predicted by Rosetta with the energy function from Barlow et al.[30], (iii) log2fc enrichment scores of mutations of sACE2 within the 23-mer sequence experimentally determined by Procko[29]. All binding affinity scores have been re-scaled to 0 (highest) to 1 (lowest) for visualization. Original amino acids are indicated on the y axes. **b** Analysis of RBD-sfGFP degradation by flow cytometry. All samples were performed in independent transfection duplicates (n = 2) and gated on GFP+ fluorescence. All indicated samples were co-transfected with RBD-sfGFP and TRIM21 in trans, and mean percentage of GFP+ cell depletion was calculated in comparison with the RBD-sfGFP-only control. 23-mer mutations are underlined according to origin. **c** Architecture and mechanism of CHIPΔTPR-based degradation system. CHIPΔTPR is fused to the C-terminus of RBD-targeting pepides. CHIPΔTPR can thus tag RBD-sfGFP complexes for ubiquitin-mediated degradation in the proteasome. **d** Analysis of RBD-sfGFP degradation by flow cytometry in the presence of Fc (in cis) and TRIM21 (in trans) or CHIPΔTPR (in cis). All samples were performed in independent transfection duplicates (n = 2) and gated on GFP+ fluorescence. Mean percentage of GFP+ cell depletion was calculated in comparison with the RBD-sfGFP only control.

Finally, numerous previous works have attempted to redirect E3 ubiquitin ligases by replacing their natural protein binding domains with those targeting specific proteins[31–33]. In 2014, Portnoff, et al.[34] reprogrammed the substrate specificity of a modular human E3 ubiquitin ligase called CHIP (carboxyl-terminus of Hsc70-interacting protein) by replacing its natural substrate-binding domain with designer binding proteins to generate optimized "ubiquibodies" or uAbs. To engineer a single construct that can mediate SARS-CoV-2 degradation without the need for trans expression of TRIM21, we fused the RBD-binding proteins to the CHIPΔTPR modified E3 ubiquitin ligase domain (Fig. 3c). After co-transfection in HEK293T cells with the RBD-sfGFP complex, we observed that the 23-mer (A2N) mutant peptide maintained equivalent levels of degradation between the TRIM21 and CHIPΔTPR fusion architecture, and was more potent than that of sACE2, sACE2v2.4, and the original 23-mer (Fig. 3d). Full-length sACE2 and optimized mutant sACE2v2.4, in particular, were less compatible with the CHIPΔTPR fusion architecture, possibly owing to steric occlusion caused by the larger size of the initial complex.

**Inhibition of infection-competent viral production.** We sought to assess the efficacy of our 23-mer (A2N)-CHIPΔTPR fusion against viruses pseudotyped with the SARS-CoV-2 S protein. We introduced a plasmid encoding our construct during lentiviral production with a ZsGreen expression plasmid, lentivirus packaging plasmid, and an envelope protein plasmid encoding the full-length S protein, rather than just the RBD (Fig. 4a). After viral supernatant recovery, we infected HEK293T cells expressing doxycycline-induced hACE2, and quantified infection as the percentage of ZsGreen+ cells by flow cytometry. Our data show that our 23-mer (A2N)-CHIPΔTPR fusion reduces the infection rate of the pseudovirus by ~60%, in agreement with our RBD-sfGFP degradation data (Fig. 4b).

Finally, we wished to test other possible mechanisms of action for our peptide variant, namely competitive interception of S protein-pseudotyped virus prior to cellular entry. To do this, we synthesized the 23-mer (A2N) peptide, and repeated our pseudoviral assay in its presence or absence at a standard dosage (1 μg/ml)[14]. We observed minimal difference in pseudoviral infection competency with the addition of the exogenous peptide in all tested experimental conditions, thus suggesting that intracellular delivery of our 23-mer (A2N)-CHIPΔTPR fusion may be the optimal modality for the peptide to inhibit viral infection (Fig. 4c).

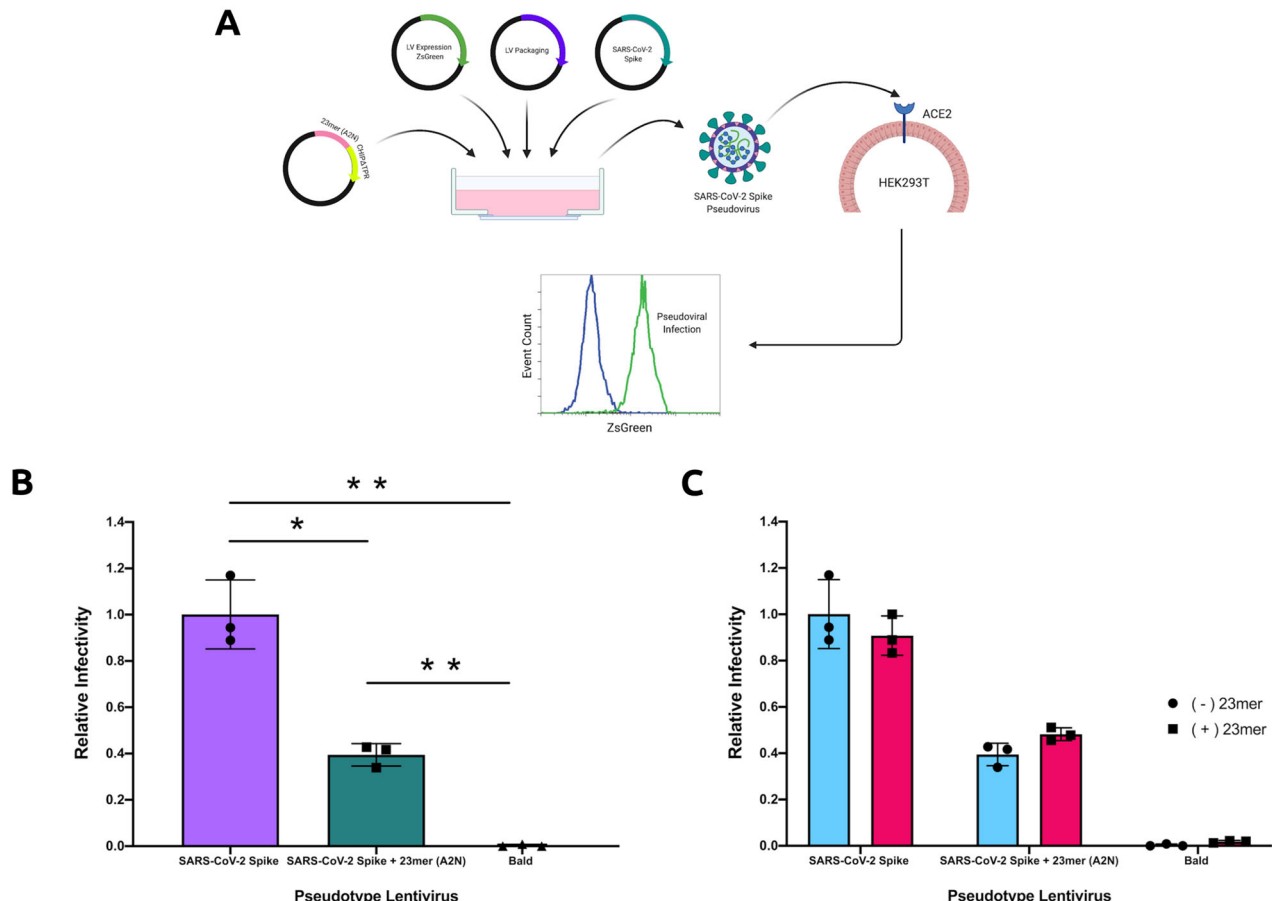

**Fig. 4 Inhibition of infection-competent viral production. a** Schematic demonstrating repurposed pseudoviral infection assay. Pseudotyped lentivirus is produced with a ZsGreen expression plasmid, lentivirus packaging plasmid, and an envelope protein plasmid encoding the S protein, in the presence of the (A2N)-CHIPΔTPR fusion plasmid. After viral supernatant recovery, hACE2 expressing HEK293T cells are infected, and infection rate is calculated as the percentage of ZsGreen+ cells by flow cyometry. **b** Analysis of pseudoviral infection inhibition by flow cytometry. All samples were performed in independent transduction triplicates ($n = 3$) and gated on ZsGreen+ fluorescence. Relative infectivity ratios were calculated by normalizing ZsGreen+ fluorescence percentages to the SARS-CoV-2 Spike (induced ACE2) positive control. Standard deviation was used to calculate error bars. Statistical analysis was performed using the two-tailed Studentailed Student's *t* test, using the GraphPad software package. Calculated *P* values are represented as follows: $^{*}P \leq 0.05$, $^{**}P \leq 0.01$. **c** Analysis of pseudoviral infection inhibition with or without exogenous addition of peptide by flow cytometry. All samples were performed in transduction triplicates ($n = 3$) and gated on ZsGreen+ fluorescence. Relative infectivity ratios were calculated by normalizing ZsGreen+ fluorescence percentages to the SARS-CoV-2 Spike (induced ACE2) positive control. Standard deviation was used to calculate error bars.

## Discussion

In this study, we have computationally truncated and engineered the human ACE2 receptor sequence to potently bind to the SARS-CoV-2 RBD. We have further identified an optimized peptide variant that enables robust degradation of RBD-sfGFP complexes in human cells, both in trans and in cis with human E3 ubiquitin ligases. Finally, we have shown that our optimal fusion construct inhibits the production of infection-competent viruses pseudotyped with the full-length S protein of SARS-CoV-2.

Although further testing contexts are needed, there may be certain advantages to our platform as compared to the PAC-MAN strategy presented recently[6], apart from the ethical implications of applying CRISPR in humans[35]. First, both of the peptide and E3 ubiquitin ligase components have been engineered from endogenous human proteins, unlike Cas13d, which is derived from *Ruminococcus flavefaciens* bacteria, thus potentially reducing the risk of immunogenicity. In addition, in terms of in vivo delivery as RNA or recombinant protein, Cas13d has an open-reading frame (ORF) of nearly 1000 amino acids, not including the guide RNAs needed for interference. The entire peptide-CHIPΔTPR ORF consists of just over 200 amino acids, which can be readily synthesized as a peptide or be efficiently

packaged for delivery in a lipid nanoparticle or adeno-associated virus.

Furthermore, our peptide fusion platform as a prophylactic provides a viable alternative to current antiviral strategies being explored for COVID-19. Antiretroviral protease inhibitors for HIV, such as lopinavir and ritonavir, have shown minimal efficacy in clinical trials of COVID-19, and generated adverse effects in a subsection of patients[36]. Similarly, antimalarials, such as hydroxychloroquine and chloroquine, which may glycosylate ACE2, have demonstrated no benefit in patients infected with SARS-CoV-2 in randomized, controlled studies[37]. Finally, there is a global effort to generate a vaccine for COVID-19. Although 11 candidates have advanced into Phase 3 trials and 5 have been approved for early or limited use in China and Russia, a standard timeframe to fully assess safety and efficacy takes well over one year[4]. Though our platform likely requires synthesis and assessment of a gene therapy, rather than a small molecule or compound, and does not generate immunological memory against SARS-CoV-2 as would a vaccine, its rapid and direct targeting mechanism, coupled with its size and human-protein derivation, presents numerous advantages as compared with existing strategies.

In total, we envision that the strategy of utilizing a computationally designed peptide binder linked to an E3 ubiquitin ligase can be explored not only for SARS-CoV-2, but also for other viruses and drug targets that have known binding partners. With already over 30,000 co-crystal structures currently in the PDB, and structure determination becoming more routine with advances in cryogenic electron microscopy, the computational peptide engineering pipeline presented here provides a versatile new therapeutic platform in the fight against COVID-19, future emergent viral threats, and numerous diseases.

## Methods

**Computational peptide design pipeline**. PDB Structure 6M0J containing the crystal structure of SARS-CoV-2 spike RBD bound with ACE2 was retrieved[20]. The PeptiDerive protocol[21] in the Rosetta protein modeling software[22] was used to determine the linear peptide segments between 10 and 150 amino acids with significant binding energy compared with that of the whole SARS-CoV-2-ACE2-RBD interaction. To analyze the conformational entropy of the peptide segments in the binding pocket, a combination of FlexPepDock and protein–protein docking protocols[23] in Rosetta was used to dock the peptides to the original RBD. All peptides were placed in the binding pocket of SARS-CoV-2 RBD for local docking. Using FlexPepDock, 300 models were created for each peptide and the top 15 models were selected to calculate the score. The peptides with the best binding energies were docked against SARS-CoV RBD bound with ACE2 using PDB 2AJF[24] containing the crystal structure of SARS-CoV spike RBD bound with ACE2. Peptides that demonstrated highest binding energy for SARS-CoV and SARS-CoV-2 RBD were docked against the $\alpha5\beta1$ integrin ectodomain using the crystal structure provided by PDB 3VI4[25].

The 23-mer peptide that showed high experimental binding affinity was selected for computational mutagenesis. A complete single point mutational scan was run for all 23 positions in the peptide using ddG-backrub script in Rosetta. For each mutation, 30,000 backrub trials were run to sample conformational diversity. The top eight mutations predicted by this protocol were chosen for the experimental assay, along with the top 8 mutations predicted using an Rosetta energy function optimized for predicting the effect of mutations on protein–protein binding[30], as well as the top 8 mutational sites predicted by Procko on sACE2[29].

**Generation of plasmids**. pcDNA3-SARS-CoV-2-S-RBD-sfGFP (Addgene #141184) and pcDNA3-SARS-CoV-2-S-RBD-Fc (Addgene #141183) were obtained as gifts from Erik Procko. hACE2 (Addgene #1786) was obtained as a gift from Hyeryun Choe. Respective peptide DNA coding sequences (CDS) were amplified from hACE2 via PCR and inserted using HiFi DNA Assembly Master Mix (NEB) for Gibson Assembly into the pcDNA3-SARS-CoV-2-S-RBD-Fc backbone linearized by digestion with NheI and BamHI. pLVX puro TRIM21-GFP (Addgene #1786) was obtained as a gift from Gaudenz Danuser. The TRIM21 CDS was amplified with overhangs for Gibson Assembly-mediated insertion into the pcDNA3-SARS-CoV-2-S-RBD-Fc backbone linearized by digestion with NheI and XhoI. pcDNA3-R4-uAb (Addgene #101800) was obtained as a gift from Matthew DeLisa. Candidate sACE2 sequences were amplified from hACE2 with overhangs for Gibson Assembly-mediated insertion into linearized pcDNA3-R4-uAb digested with HindIII and EcoRI. Single amino-acid substitutions were introduced utilizing the KLD Enzyme Mix (NEB) following PCR amplification with mutagenic primers (Genewiz). Assembled constructs were transformed into 50 μL NEB Turbo Competent *Escherichia coli* cells, and plated onto LB agar supplemented with the appropriate antibiotic for subsequent sequence verification of colonies and plasmid purification.

**Cell culture and flow cytometry analysis**. HEK293T cells were maintained in Dulbecco's Modified Eagle's Medium supplemented with 100 units/ml penicillin, 100 mg/ml streptomycin, and 10% fetal bovine serum (FBS). RBD-sfGFP (333 ng),

peptide (333 ng), and E3 ubiquitin ligase (333 ng) plasmids were transfected into cells as duplicates ($2 \times 10^5$/well in a 24-well plate) with Lipofectamine 3000 (Invitrogen) in Opti-MEM (Gibco). For transfection conditions with fewer required plasmids, we co-transfected cells with the necessary amounts of empty pcDNA3.1 plasmid to obtain 999 ng of total DNA. After 5 days post transfection, cells were harvested and analyzed on an Attune® NxT Flow Cytometer (Thermo Fisher) for GFP fluorescence (488-nm laser excitation, 530/30 filter for detection). Cells expressing GFP were gated, and percent GFP+ depletion to the RBD-sfGFP only control were calculated. All samples were performed in independent transfection duplicates ($n = 2$), and percentage depletion values were averaged. Standard deviation was used to calculate error bars.

**Cell line generation**. Doxycycline-inducible hACE2 HEK293T/17 cells were made by co-transfection in a six-well plate of 2 μg Piggy-Bac expression plasmid containing a doxycycline-inducible hACE2 gene and puromycin selection marker and 0.5 μg of a Super PiggyBac Transposase expression vector. After 6 days of 3 μg/ml puromycin selection, the cells were harvested and genomic DNA was extracted for diagnostic PCR to confirm integration of the expression cassette. In addition, mRNA was extracted 24 hours post doxycycline induction for use in qPCR to measure hACE2 expression levels.

**Pseudovirus infection assay**. Pseudotyped lentivirus was produced according to Thermo Fisher Lipofectamine 3000 lentivirus production protocol. In brief, HEK293T/17 cells were seeded at high density in 10 cm dishes in lentivirus packaging media (Gibco Opti-Mem Reduced serum medium and 5% heat inactivated FBS) 24 hours prior to transfection. All lentiviruses utilized 7 μg of CMV-ZsGreen lentivirus expression plasmid (Addgene #124301) as well as 7 μg of the second generation lentivirus packaging plasmid psPAX2 (Addgene #12260). For envelope proteins, 4 μg of a positive control VSV-G envelope plasmid was used to make a VSV-G pseudotyped LV, 4 μg of the Promega Transfection carrier DNA was used instead of envelope protein to make the Bald pseudotyped lentivirus, and 4 μg of the SARS-CoV-2 Spike protein (Addgene #145032) was used to make SARS-CoV-2 Spike lentivirus. To make the SARS-CoV-2 Spike + 23-mer (A2N) pseudotyped lentivirus, 4 μg of the 23-mer (A2N) plasmid and 4 μg of SARS-Cov-2 Spike were added. After mixing the plasmids with Lipofectamine 3000, P3000 reagent in Opti-Mem, HEK293T cells were transfected and the media was changed after 6 hours. Lentivirus-infused media was then collected at 24 hours post transfection and 52 hours post transfection. The virus-containing media was then centrifuged at 2000 rpm and the supernatant was filtered through a 45 μm filter to remove cellular debris. The 24 ml of filtered viral supernatant was then further concentrated to 1 ml using the Lenti-X viral concentrator by adding 3× Lenti-X, mixing, incubating for 45 minutes at 4 °C, then centrifuging for 45 mins at 1500 *g*. The viral pellet was then resuspended in 1 ml of PBS and aliquoted for use in transduction.

Doxycycline-inducible hACE2 HEK293T/17 cells were made by transfection of a PiggyBac expression plasmid containing a doxycycline-inducible hACE2 gene and a puromycin selection marker and a PiggyBac transposase vector. After 6 days of 3 μg/ml puromycin selection, the cells were harvested and gDNA was extracted for diagnostic PCR to confirm integration of the expression cassette. In addition, mRNA was extracted 24 hours post doxycycline induction for use in qPCR to measure hACE2 expression levels.

All viral transduction was performed in 24-well culture dishes. In brief, control HEK293T/17s, non doxycycline-induced hACE2 293T/17s and Dox-induced hACE2 293T/17s were plated at high density and were 50% confluent at the time of transduction. Viral aliquots were thawed for 2 minutes at 37 °C prior to use and were not subjected to multiple freeze–thaw cycles. In triplicate, 40 μL of concentrated virus was added to each well of the 293Ts in media containing 5 μg/ml polybrene. In all, 2 μg/ml of doxycycline was added along with the virus. For the peptide blocking assay, 1 μg/ml of synthesized peptide (Amide Technologies) was used for blocking, and pseudovirus (with or without peptide) was incubated for 1 hour at 37 °C prior to transduction. After 36 hours, cells were harvested for flow cytometry and percentage of ZsGreen+ cells was calculated in each condition, according to the gating strategy highlighted in Fig. 5. Because of low number of

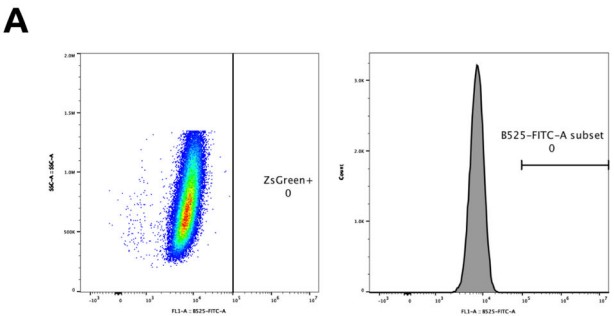

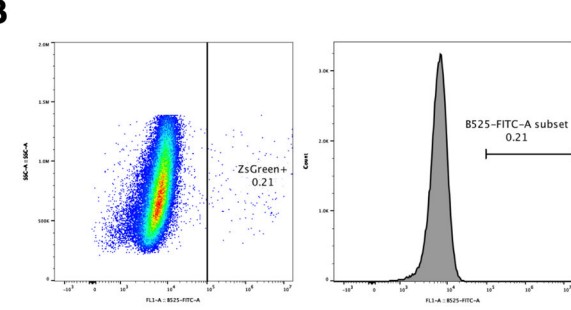

**Fig. 5 Flow cytometry gating strategy. a** Negative control to establish ZsGreen+ gate. **b** Evaluation of gate on cell samples expressing ZsGreen.

ZsGreen+ cells, due to low titer virus and likely lack of viral co-receptor expression in HEK293T cells, relative infectivity ratios were calculated by normalizing ZsGreen+ fluorescence percentages to the SARS-CoV-2 Spike (induced ACE2) positive control. Statistical analyses were performed using the two-tailed Student's *t* test, using the GraphPad software package. Calculated *P* values are represented as follows: [*]$P \leq 0.05$, [**]$P \leq 0.01$.

**Statistics and reproducibility**. All samples were performed in independent transfection duplicates ($n = 2$) or triplicates ($n = 3$), as indicated, and percentage depletion values were averaged. Standard deviation was used to calculate error bars. Statistical analyses was performed using the two-tailed Student's *t* test, using the GraphPad software package. Calculated *P* values are represented as follows: [*]$P \leq 0.05$, [**]$P \leq 0.01$.

**Reporting summary**. Further information on research design is available in the Nature Research Reporting Summary linked to this article.

## Data availability
All data needed to evaluate the conclusions in the paper are present in the paper. All source computational and experimental data files can be accessed at: https://doi.org/10.5281/zenodo.4151380.

## Code availability
All source code described in this project can be accessed at: https://doi.org/10.5281/zenodo.4154988.

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

## Acknowledgements
We thank Dr. Neil Gershenfeld and Dr. Shuguang Zhang for shared lab equipment, and Amide Technologies for peptide synthesis. We further thank Eyal Perry for computational assistance, and James Weis and Nest.Bio Labs for access to flow cytometry. This work was supported by the consortia of sponsors of the MIT Media Lab and the MIT Center for Bits and Atoms, and by Jeremy and Joyce Wertheimer.

## Author contributions
P.C. conceived, designed, and supervised the study. P.C. designed and built constructs, carried out experiments, and conducted data analyses. M.P. implemented computational design pipelines and performed in silico docking protocols. A.M.P. engineered hACE2 cell line, performed viral transduction, and conducted flow cytometry on transduced cells. C.K. produced lentiviral pseudotypes, engineered hACE2 cell line, and performed transduction protocols. P.C. wrote the paper, with input from M.P., C.K., and A.M.P. G.M.C. supervised pseudovirus work. J.M.J. reviewed the paper and provided critical insight and ideas.

## Competing interests
P.C. and J.M.J. are listed inventors for provisional patent application entitled "Minimal Peptide Fusions for Targeted Intracellular Protein Degradation." The remaining authors declare no competing interests.
