## [Peer Review File · Communications Biology]

Reviewers' Comments:

Reviewer #1:

Remarks to the Author:

In this manuscript, the authors describe the computational design and in vitro test of ACE2-derived peptides that target the SARS-CoV-2 RBD and E2 ubiquitin ligase as a putative antiviral strategy.

Overall, the authors' approach is novel, with an interesting design that could be tailored to mutations that may arise in SARS-CoV-2 once vaccinations begin to become available. With that in mind, this manuscript would be strengthened by the authors commenting on the tailorability of this strategy, both for SARS-CoV-2 in specific and in the event other coronaviruses emerge.

In addition, some discussion of the strengths/weaknesses of this strategy for prophylaxis versus other strategies (beyond the CRISPR-Cas13d strategy mentioned) would be warranted.

Reviewer #2:

Remarks to the Author:

In the manuscript by Chatterjee et al., the authors use computationally-designed ACE2-derived peptides to redirect the SARS-CoV-2 S to E3 ubiquitin ligases for intracellular degradation. Though there are other papers that also use soluble ACE2 or soluble ACE2 variants as binding partners to neutralize SARS-CoV-2 infections (Chan et al., Science, 2020), the combination of soluble ACE2 binding with E3 ubiquitin ligases is quite new. The manuscript is concise and easy to understand. The method section is thorough and should be easy to follow by other researchers.

I have the following comments/suggestions:

1. In Fig.4, the authors evaluated the efficacy of the 23 mer-CHIP- Δ TPR in pseudovirus expressing SARS-CoV-2 S protein and concluded that the peptide can inhibit the infection. However, this inhibition may be at the pre-entry level, in which the 23 mer peptide out-competes the cellular ACE2 receptors for binding with S and results in less virus entry. A control experiment using 23 mer peptide without-chip- Δ TPR may inform on whether this is caused at the pre-entry level.
2. Authors claim that the peptide can target the viral spike protein for the intracellular degradation of SARS-CoV-2 in the proteasome. However, S-RBD-sfGFP was used to validate the efficacy. I appreciate that the authors mention that validation should be made in the context of real virus. It would be good if the authors can try in full-length S.
3. In Fig.2C, it would be great to include a control of peptide + TRIM21, as this would inform the global change caused by the overexpression of TRIM21. It also lays the foundation for the data presented in Fig. 3.
4. I think at least three independent transfections/experiments should be done to come up with a statistically significant conclusion.
5. Authors use a diagram in Figs 2 and 3 to illustrate the mechanism. This helps audience to understand the story. However, it is a bit misleading when the diagram is presented before the data. It seems like real viruses are used for the experiments. It would be helpful if the diagram follows the data.
6. In Fig 3D, data show the patterns of sACE2 mediated TRIM and CHIP- Δ TPR degradation are quite different, while 23mer mediated TRIM and CHIP- Δ TPR degradation are similar. Can authors provide an explanation?
7. Please use bigger font on text in figures (e.g. axis title).

Reviewer #3:

Remarks to the Author:

In the manuscript "Target Intracellular Degradation of SARS-CoV-2 via Computationally-Optimized Peptide Fusions", the authors performed a computational-based strategy by engineering optimized ACE2-derived peptides that are able to simultaneously target the Spike RBD and the human E3 ubiquitin ligases for intracellular degradation of the virus. In this way, they hypothesized that such antiviral strategy may potentially inhibit infection-competent viral production.

Authors supported their major hypothesis using various software and computational methods, along with an in vitro pseudovirus infection assay. They design a novel therapeutic platform that may have clinical applicability for the future design of peptide entry/fusion inhibitors.

This is certainly a valuable approach that has the potential to provide new insights compared to the cited PAC-MAN strategy previously described by Abbott et al. (2020).

Manuscript is well written and results are well interpreted and discussed.

In order to further improve the quality of this manuscript, I would suggest the authors to make the following modifications:

1) At page 2, lines 17 to 21: An update of the data is suggested when the review is made.

2) At page 2, Line 23: "With no vaccine or cure readily available". Although no vaccine is widely available, five vaccines were approved for early or limited use (four approved for limited or early use in China and one approved for early use in specific populations in Russia).

3) At page 2, lines 29 to 32: The use of CRISPR-Cas13d or similar genome editing technologies in human cells, presents delivery and safety challenges but also, and very important, ethical implications that were not acknowledge and could be addressed in the discussion and/or mentioned in any other part of the manuscript.

4) At page 2, Line 36: "Thus, we hypothesize that (...) one can mediate depletion of SARS-CoV-2 in vivo". Since the authors only performed in silico and in vitro assays, they should be more cautious in addressing such hypothesis as introducing their main goals. It is suggested to rephrase this sentence or to better contextualize the objective in the experiments actually carried out.

5) At page 4, Line 64: "Furthermore, it is critical that therapeutics targeting SARS-CoV-2 epitopes withstand the possibility of viral mutation (...) We conducted in silico protein modeling to engineer minimal sACE2 peptides that not only maintain potent RBD binding (...) and exhibit cross-binding affinity toward the previous SARS-CoV spike protein".

A major concern in this case is highlighted by the apparent disregard of the spike protein conservation across SARS-CoV and SARS-CoV-2. I suggest the authors to consider and discuss the implications of the RBD conservation, specifically, the binding pocket for the SARS-CoV-2 and include the relevant information in the revised manuscript. The reading of the following articles could be useful (Cagliani et al (2020); Wall et al (2020); Trigueiro-Louro et al (2020)).

6) At page 4, Line 76 and at page 8, line 178: "SARS-CoV-2-sACE2-RBD" should be probably changed to "SARS-CoV-2 RBD-sACE2"

7) Material and Methods: The selected structures 6M0J and 2AJF contain the Spike RBD bound

with the ACE2. Since bounded ligands may alter the native conformation of proteins; and considering the in silico studies were only performed in ACE2-bound RBD structures, this question should be clearly indicated and clarified.

Point-By-Point Response to Reviewers

COMMSBIO-20-2400

Reviewer #1 (Remarks to the Author):

In this manuscript, the authors describe the computational design and in vitro test of ACE2-derived peptides that target the SARS-CoV-2 RBD and E2 ubiquitin ligase as a putative antiviral strategy.

Overall, the authors' approach is novel, with an interesting design that could be tailored to mutations that may arise in SARS-CoV-2 once vaccinations begin to become available. With that in mind, this manuscript would be strengthened by the authors commenting on the tailorability of this strategy, both for SARS-CoV-2 in specific and in the event other coronaviruses emerge.

In addition, some discussion of the strengths/weaknesses of this strategy for prophylaxis versus other strategies (beyond the CRISPR-Cas13d strategy mentioned) would be warranted.

We thank the reviewer for the constructive feedback on how to improve the manuscript. In the Results and Discussion section, we concisely add sections on RBD conservation as well as tailorability of the peptide fusion strategy for other biological targets. Finally, we add a short paragraph in the Discussion about the strengths and weaknesses of our system versus other prophylactic strategies, such as protease inhibitors and vaccines.

Reviewer #2 (Remarks to the Author):

In the manuscript by Chatterjee et al., the authors use computationally-designed ACE2-derived peptides to redirect the SARS-CoV-2 S to E3 ubiquitin ligases for intracellular degradation. Though there are other papers that also use soluble ACE2 or soluble ACE2 variants as binding partners to neutralize SARS-CoV-2 infections (Chan et al., Science, 2020), the combination of soluble ACE2 binding with E3 ubiquitin ligases is quite new. The manuscript is concise and easy to understand. The method section is thorough and should be easy to follow by other researchers.

I have the following comments/suggestions:

1. In Fig.4, the authors evaluated the efficacy of the 23 mer-CHIP- Δ TPR in pseudovirus expressing SARS-CoV-2 S protein and concluded that the peptide can inhibit the infection. However, this inhibition may be at the pre-entry level, in which the 23 mer peptide out-competes the cellular ACE2 receptors for binding with S and results in less virus entry. A control experiment using 23 mer peptide without-chip- Δ TPR may inform on whether this is caused at the pre-entry level.

We thank the reviewer for pointing out this possible mode of action of our peptide. To assay whether the peptide can inhibit infection through an extracellular interception of the pseudovirus, we repeated Figure 4 in triplicates to include the addition of peptide at a standard dosage. Our data show that the peptide does not have an effect on infection inhibition. We include this data as Figure 4C, and include a description in the Results section.

2. Authors claim that the peptide can target the viral spike protein for the intracellular degradation of SARS-CoV-2 in the proteasome. However, S-RBD-sfGFP was used to validate the efficacy. I appreciate that the authors mention that validation should be made in the context of real virus. It would be good if the authors can try in full-length S.

The reviewer correctly mentions that we conducted our preliminary engineering studies with the Spike RBD, rather than the real virus, and suggested that we test our system with full-length Spike (S) protein. However, in Figure 4 (which we have updated according to point 1 and 4 of the reviewer's comments), we conducted a pseudovirus infection assay with the full-length Spike protein, as this was the main experiment to validate our system's efficacy. We believe that this data demonstrates the ability of our peptide to target full-length S protein.

3. In Fig.2C, it would be great to include a control of peptide + TRIM21, as this would inform the global change caused by the overexpression of TRIM21. It also lays the foundation for the data presented in Fig. 3.

We apologize to the reviewer for not including this data in the original graph. In the previous version of the manuscript, in line 102 of our manuscript, we indicate that we did run the control of just over expressing TRIM21 in the presence of each binder, and we include the data in our raw datasets, but we did not place the data in the figure (to avoid over cluttering the chart). Rather than repeating the experiment, we added that data (conducted in the same experimental run) to the figure to satisfy the reviewer's request.

4. I think at least three independent transfections/experiments should be done to come up with a statistically significant conclusion.

As the pseudovirus assay is the core experiment to prove out system's efficacy, we reconducted this experiment in triplicates (Figure 4B), and included the exogenous peptide experiment (from Point 1) within the figure (Figure 4C). We hope that this satisfies the reviewer's request, without having to reperform all of the experiments in the paper, as the previous experiments (in Figure 2 and 3) were done to simply engineer an optimal architecture.

5. Authors use a diagram in Figs 2 and 3 to illustrate the mechanism. This helps audience to understand the story. However, it is a bit misleading when the diagram is presented before the data. It seems like real viruses are used for the experiments. It would be helpful if the diagram follows the data.

We thank the reviewer for this very appropriate suggestion. We have updated both mechanism figures to show the peptide fusions targeting an RBD-sfGFP complex, rather than the virus itself.

6. In Fig 3D, data show the patterns of sACE2 mediated TRIM and CHIP- Δ TPR degradation are quite different, while 23mer mediated TRIM and CHIP- Δ TPR degradation are similar. Can authors provide an explanation?

We appreciate the reviewer pointing out this phenomenon, as we did not include an explanation in the original manuscript. After internal discussions with authors and collaborators, we have updated the manuscript text to add a short explanation in the results section for Figure 3D. We hope this explanation is satisfactory for the reviewer.

7. Please use bigger font on text in figures (e.g. axis title).

We have made the font in the figures as big as possible with the software we are using. We hope that the figures are now more readable for the reviewer.

Reviewer #3 (Remarks to the Author):

In the manuscript "Target Intracellular Degradation of SARS-CoV-2 via Computationally-Optimized Peptide Fusions", the authors performed a computational-based strategy by engineering optimized ACE2-derived peptides that are able to simultaneously target the Spike RBD and the human E3 ubiquitin ligases for intracellular degradation of the virus. In this way, they hypothesized that such antiviral strategy may potentially inhibit infection-competent viral production.

Authors supported their major hypothesis using various software and computational methods, along with an in vitro pseudovirus infection assay. They design a novel therapeutic platform that may have clinical applicability for the future design of peptide entry/fusion inhibitors.

This is certainly a valuable approach that has the potential to provide new insights compared to the cited PAC-MAN strategy previously described by Abbott et al. (2020).

Manuscript is well written and results are well interpreted and discussed.

In order to further improve the quality of this manuscript, I would suggest the authors to make the following modifications:

1) At page 2, lines 17 to 21: An update of the data is suggested when the review is made.

We have made the update as of October 1, 2020.

2) At page 2, Line 23: “With no vaccine or cure readily available”. Although no vaccine is widely available, five vaccines were approved for early or limited use (four approved for limited or early use in China and one approved for early use in specific populations in Russia).

We thank the reviewer for pointing this omission out. We have updated the text in both the introduction and discussion to indicate the approval of these early vaccines.

3) At page 2, lines 29 to 32: The use of CRISPR-Cas13d or similar genome editing technologies in human cells, presents delivery and safety challenges but also, and very important, ethical implications that were not acknowledge and could be addressed in the discussion and/or mentioned in any other part of the manuscript.

The reviewer makes a very good point that the use of CRISPR in humans is potentially ethically problematic. However, we would like to point out that CRISPR-Cas13d is an RNA-targeting system, which does not have the ability to make permanent changes to the genome, rather just to destroy transiently-expressed RNA molecules. Thus, the same ethical problems with CRISPR-Cas9 or CRISPR-Cas12 would not apply in this situation. We do, however, make a small note of CRISPR ethics in the text.

4) At page 2, Line 36: “Thus, we hypothesize that (...) one can mediate depletion of SARS-CoV-2 in vivo”. Since the authors only performed in silico and in vitro assays, they should be more cautious in addressing such hypothesis as introducing their main goals. It is suggested to rephrase this sentence or to better contextualize the objective in the experiments actually carried out.

We have rephrased the sentence to be: “Thus, we hypothesize that by employing guiding E3 ubiquitin ligases with viral-targeting peptides, one can mediate depletion of SARS-CoV-2 viral components in vitro.” We hope this satisfies the reviewer’s concerns.

5) At page 4, Line 64: “Furthermore, it is critical that therapeutics targeting SARS-CoV-2 epitopes withstand the possibility of viral mutation (...) We conducted in silico protein modeling to engineer minimal sACE2 peptides that not only maintain potent RBD binding (...) and exhibit cross-binding affinity toward the previous SARS-CoV spike protein”.

A major concern in this case is highlighted by the apparent disregard of the spike protein conservation across SARS-CoV and SARS-CoV-2. I suggest the authors to consider and discuss the implications of the RBD conservation, specifically, the binding pocket for the SARS-CoV-2 and include the relevant information in the revised manuscript. The reading of the following articles could be useful (Cagliani et al (2020); Wall et al (2020); Trigueiro-Louro et al (2020)).

We thank the reviewer for this very important point and the helpful suggested reading. We have included a paragraph on RBD conservation within the computational design section of the Results to indicate the high conservation of the RBD, in accordance with the data reported in the three papers. We have further added the citations to the references list.

6) At page 4, Line 76 and at page 8, line 178: “SARS-CoV-2-sACE2-RBD” should be probably changed to “SARS-CoV-2 RBD-sACE2”

We have made this correction in the text.

7) Material and Methods: The selected structures 6M0J and 2AJF contain the Spike RBD bound with the ACE2. Since bounded ligands may alter the native conformation of proteins; and considering the *in silico* studies were only performed in ACE2-bound RBD structures, this question should be clearly indicated and clarified.

We agree with the reviewer that it’s critical to assess the conformational stability of the peptides in the unbound state. We completed those *in silico* experiments when designing the peptides and have now indicated this in the Results section.

Reviewers' Comments:

Reviewer #2:

Remarks to the Author:

I support the publication of this manuscript.

Reviewer #3:

Remarks to the Author:

The manuscript "Target Intracellular Degradation of SARS-CoV-2 via Computationally-Optimized Peptide Fusions", was improved by the authors through the inclusion of the suggestions and recommendations that were made. The authors took good note of the issues raised and modified the manuscript accordingly.